# Unveiling the obstacles encountered by women doctors in the Pakistani healthcare system: A qualitative investigation

**Ali Raza**[1]*, **Junaimah Jauhar**[2], **Noor Fareen Abdul Rahim**[3], **Ubedullah Memon**[1], **Sheema Matloob**[4]

1 Department of Business Administration, Sukkur IBA University, Sukkur, Pakistan, 2 CENTER of Excellence for Continuous Education & Development (CECED), Universiti Sains Malaysia, Penang, Malaysia, 3 Graduate Business School, Universiti Sains Malaysia, Penang, Malaysia, 4 School of Management, Universiti Sains Malaysia, Penang, Malaysia

* ali.raza@iba-suk.edu.pk

**Data Availability Statement:** All relevant data are within the paper.

## Abstract

In Pakistan, women outnumber men in medical colleges with 80 percent enrollment, yet many fail to practice medicine following graduation. Pakistan Medical Council (PMC) states 50 percent of graduated women doctors either did not practice or left employment in a short period. Thus, the non-servicing women doctors are assumed as the one of the major causes for the overall doctors' shortage in the country. Addressing this enduring matter, this study aims to explore and understand the factors that discourage women doctors from practicing medicine in Pakistani hospitals. The study employed qualitative exploratory inquiry with an interpretive paradigm to attain a deeper understanding of the problem. 59-semi structured interviews were conducted by non-working women doctors across the entirety of Pakistan. The narratives were then analyzed by thematic analysis using ATLAS.ti 22. The findings have resulted in the three major themes, i.e., workplace challenges, socio-cultural obstructions, and familial restrictions that possibly obstruct women from practicing medicine in hospitals. The findings suggested that accepting traditional cultural values, including entrenched gender roles in society, deters women from practicing medicine. The prevailing patriarchal societal system includes stereotypes against working women; early marriages hinder women from practicing medicine. The prevailing societal system upholds the influence of in-laws and a husband for women doctor professional employment. As a result, severe work-life conflict was reported where most women doctors ended up in their profession in the middle of struggling between socially rooted gender roles as homemakers and their professional careers—furthermore, the study found various workplace issues that posit an additional burden on already struggling women doctors. Issues include poor recruitment and selection process, transfer constraints, excessive workload with inadequate salary, harassment, gender discrimination, unsafe work environment, and little support from the administration highly contribute to the shortage of women doctors in Pakistan.

**Funding:** The author(s) received no specific funding for this work.

**Competing interests:** The authors have declared that no competing interests exist.

## 1. Introduction

The 17.4 million healthcare workers shortage is hurtling global health workforce crises (World Health Organization, [WHO], 2016). WHO projects a 10 million health worker shortage by 2030, with the majority of these countries being low- and lower-middle income nations [1]. Undoubtedly, health systems can only function with healthcare workers; and the ongoing health workforce crisis primarily affects low and middle-income countries, particularly countries in South Asian and African regions, where the burden of diseases [BoD] is high where there are severe disruptions reported in the expansion of healthcare coverage, standards in terms of availability, accessibility, acceptance, and quality. The shortfall of health workers affects developing countries like Pakistan the most. WHO registered Pakistan as one of the 57 countries with a critical health workforce deficiency and indicated that Pakistan's poor workforce policy is a significant constraint to the availability of adequate human resources for health [2]. In addition, the prevailing severe health workforce crisis in Pakistan has seen as the primary reason for the overall healthcare delivery system's deterioration in the country [3].

Presently, the Pakistan health sector comprises 123 medical undergraduate institutions schools, 48 public, and 75 private [4], which produces approximately 12,000 to 14,000 doctors annually. At the moment, there are 271,560 medical doctors registered with PMC; among them, 127,468 (46.9%) are females, whereas 144,092 (53.1%) are male doctors [4, 5]. These registered doctors are considered ready for practice in different capacities across the country's health institutions. However, this number does not seem sufficient to cater to the healthcare needs of 225.5 million populous countries. Based on the international standard of 2 physicians per 1,000 population, it is concluded that there is shortage of approximate more than 178,000 medical doctors approximately against the need for 450,000 approximately medical doctors in the country.

PMC claims that one of the major causes of the overall medical doctor shortage is non-practicing women doctors in Pakistan. PMC statistics shows that women enrollment in medical schools significantly outnumbered their male counterparts by 80 percent, yet half, i.e., 50 percent of the women medical graduates, either fail to join the workforce or discontinue their jobs early in their careers [5–7] Here, it is essential to mention that the country-wide shortage of more than 178,000 doctors was determined by the number of doctors registered with PMC. More seriously, the registered doctors do not represent the actual number of practicing doctors in hospitals. In this way, among 127,468 women doctors registered, only 63,734 are assumed practicing in hospitals, which is a negligible proportion. This phenomenon shows that increase in women medical graduates has not translated into an actual increase in the number of practicing doctors [8, 9]. In the same way, approximately 85,000 women doctors are out of practice in Pakistan [10].

The state authorities and the healthcare department consider the predicament not only as a significant waste of valuable resources but also as the quality of overall healthcare delivery. Financially, the state-run medical charges a nominal fee of approximately $1,000 for a complete five-year MBBS degree, whereas estimation shows that it requires around $25,000 to educate a single medical student till graduation [6, 7]. Despite these measures, there are only 50,000 to 60,000 physicians present at hospitals against the demand of 450,000, and the shortage of 178,440 doctors prompted a grave concern for the health authorities.

The effects of health workforce crises can also be observed while viewing Pakistan's poor positioning in different health indicators. The physician-to-population ratio is as low as 1.1 per 1,000 population [1]. In addition, UNDP [11] states Pakistan's Human Development Index (HDI) value is 0.560, which positioned the nation at 152 out of 189 countries. Similarly,

WHO ranked Pakistan 122 among 191 countries in the overall quality of healthcare systems, including adequate health infrastructure and healthcare professionals. Miserably, Pakistan placed 154 out of 191 countries in the global health care access and quality index (HAQ), where the BoD is high [12]. Apart from quality health care, Pakistan's position is highest in newborn mortality rates, 1 in 22, which accounts for 61.2 per 1,000 live births [13]. Similarly, the nation's position is 149th out of 179 Maternal Mortality Ratio Index [6] compared to the same region countries such as Sri Lanka 30, China 27, Malaysia 40, and Thailand 20 [14]. Overall, Pakistan has a low life expectancy, which is 67.4 years. In contrast, China stands at 76.4, Malaysia and Thailand at 75.5, Bangladesh at 72.8, Nepal and Bhutan at 70.6.

In a country of 225.5 million, the on-going shortage of medical doctors has far-reaching implications for the community. Several stakeholders, including health officials, debaters, and media influences, have held non-practicing women doctors responsible for the country's overall shortage of healthcare workers. In 2014, PMC recommended restricting the admission of women to 50 percent to deal with the problem of doctors' shortage. However, the case is currently on hold following in challenge in the High Court of Punjab, Pakistan [15].

Here, it is important to emphasize that Pakistan is one of the countries with a massive gender disparity where women are suffering in almost all spheres of life. Pakistan ranked is highly gender un-equal country and stands at last position compared to its neighbors and other regional countries [16]. Studies have shown persistent gender disparities in educational opportunities, with limited access to enter in labor force [17]. In this context, World Bank [18] also indicated that only 20% of Pakistani women have a university degree associated with professional employment. Furthermore, Gender-based discrimination in the workplace, such as lower wages and limited career advancement for women, has been documented [19]. Women participation is recorded as low as less than 5 percent among all sectors [14]. Similarly, despite legal reforms, women continue to face challenges in exercising their legal rights due to discriminatory practices and limited access to justice [20] due to the pervasive influence of patriarchal norms on gender roles and societal expectations [19].

Based on the scenario mentioned above, the present research aims to investigate the increasing number of women doctors in medical schools but failing to practice seems paradoxical. Despite the obvious challenges women face in society, the issue caught massive attention in the media and stirred immediate controversy i.e., [21–23]. Several newspaper articles, blogs, social media posts, and television advertisements created a hype with heated discussions that concludes that women's inclination to medical careers is to only attain appropriate marriage proposals due to the "Doctor-bride Sensation" in Pakistan [24]. The community often criticized women doctors for purposively opting for medical careers for personal gain, not serving the country Nevertheless, given the challenging circumstances faced by employed women in Pakistan, it appears that this narrative may present a biased and the phenomenon is yet to be explored systemically. Considering the aforementioned situation, the present study offers a prompt diagnose the issue of the shortage of women doctors by exploring the factors that discourage women doctors to practice in in the healthcare system of Pakistan. Additionally, the study highlights the influence of traditional cultural values and deeply ingrained gender roles on women in the medical field, along with the challenges faced in the workplace. This study also offers potential practical implications that equips relevant decision-makers with the vital information required to enact crucial reforms, facilitating the reintegration of women doctors into the workforce, and harnessing their skills and potential within the field. Consequently, by addressing women-specific issues and improving working conditions, this research has the potential to positively impact the overall decline in healthcare delivery. Additionally, it can serve to amplify awareness of women's issues and facilitate smoother entry for women into the labor force across the entire nation.

## 2. Materials and methods

### 2.1. Research design and approach

Qualitative exploratory design under the interpretive epistemological assumption was deemed appropriate for the study. The interpretive approach was chosen to explore and understand the non-working women doctors about their employment decisions. Nevertheless, deeper insights cannot be explored through quantitative methods. A case study approach was used in this study [25, 26]. The case study approach facilitated an in-depth, multi-faceted understanding of non-working women doctors' personal, professional, and social lives in their real-life context to explore the phenomenon [27, 28].

### 2.2. Study participants

The participants involved women doctors who graduated from medical schools in Pakistan and possess a valid registration certificate from the PMC. These women doctors were categorized into two groups: the first group consisted of women doctors who did not practice medicine after completing their medical degree, while the second group comprised women doctors who had previously been employed but left their jobs within a period of 1 to 2 years.

### 2.3. Theoretical lens

Keeping in perspective the literature on women doctors' representation and attrition in hospitals, especially in Asian societies, "social role" theory [29] was considered as the most suitable and primary theoretical lens to explore the reasons that restrain women doctors from working in hospitals. The theory indicates that the societal system is based on strict gender roles. Thus, it restricts women from accepting socially acceptable roles; otherwise, they may encounter severe societal punishments. Deep-rooted in the social role theory [30], the present study incorporated another theoretical lens called spillover theory [31]. The theory explains that the conflict arose from work-role and family-roles. The theory can be categorized into positive and negative spillover. Negative spillover, the general explanation, is work and family conflict or interference [32, 33]. In contrast, positive spillover refers to social support, resource enhancement, and work-family success or balance [34, 35]. The present study aimed to cover both perspectives to study the life of women doctors in Pakistan.

### 2.4. Data collection instrument and procedure

The study employed a semi-structured interview approach using an interview guide, following the guidelines outlined by Cresswell and Poth [27] and Carpenter and Suto [36]. The interview guide was developed by integrating insights from a thorough review of relevant literature and ap6plying the Interview Protocol Refinement Framework (IPR) proposed by Castillo-Montoya [37]. The IPR enhances the reliability of interview guide and the quality of information recorded for the research objective. In this study, each interview offered valuable insights that were used to refine and enhance the interview guide. The information obtained from the interviews was carefully considered, leading to the addition and modification of certain questions. These adaptations were instrumental in delving deeper into the phenomenon under investigation and aligning the interview guide more closely with the perspectives and experiences of the women doctors.

 **2.4.1. Instrument.** Semi-structured interviews were the main data collection instrument. This study aimed to explore the factors that discourage women doctors from working in Pakistani hospitals which necessitated conducting in-person interviews in their natural settings. Therefore, all the interviews were conducted face-to-face to better understand the

phenomenon. Interviews was conducted using the interview guide which was developed by integrating insights from a thorough review of relevant literature and applying the Interview Protocol Refinement Framework (IPR) [37]. The IPR enhances the reliability of interview guide and the quality of information recorded for the research objective. In this study, each interview offered valuable insights that were used to refine and enhance the interview guide for future interviews. The information obtained from the interviews was carefully considered, leading to the addition and modification of certain questions. These adaptations were instrumental in delving deeper into the phenomenon under investigation and aligning the interview guide more closely with the perspectives and experiences of the women doctors. Table 1 presents the interview guide with key interview questions.

**2.4.2. Procedure.** In this study, a combination of purposive sampling and snowball sampling technique was used to recruit the women doctors. Purposive sampling facilitates in identification and selection of information-rich cases for the most effective use of limited resources [38]. This involves identifying and selecting women doctors that are especially knowledgeable about or experienced with a phenomenon of interest [27]. Likewise, the snowball sampling technique, also known as participant-driven sampling [39] was employed in this study. It involves the researcher accessing potential participants by utilizing the contact details provided by other participants in the research. By utilizing this approach, the study aimed to tap into the

**Table 1. Interview guide.**

| No. | Investigated Theme | Questions |
|---|---|---|
| 1 | Workplace Factors | In your opinion, what are the factors in a hospital setting potentially hinder the practice of women doctors? Could you please describe your experience of the challenges you faced while continuing your work? What about workload policy, progression opportunities, remunerations, and support for women doctors?<br>How do you see the process of obtaining employment in hospitals? Could you share your personal experiences in this regard?<br>In your opinion, how would you assess the suitability of the workplace culture in hospitals for women? Have you ever encountered discomfort intent or behavior while working in the hospital? When and how? How have you tackled these issues? |
| 2 | Socio-cultural Factors | To what extent are you satisfied with the status of women in a collectivist society like Pakistan? Why? As a woman, do you think you have been given an equal chance in society and family? Why?<br>What are the societal beliefs that influence Women's employment in Pakistan? Can you please share your feelings on how society sees you as a working woman, i.e., a medical doctor?<br>Can you please comment on Women's observance of the Hijab and its impact on jobs and society? Do you think Hijab ever obstructs your employment outside the home? How and why?<br>It is a typical narrative that marriage is a reason for women doctors' withdrawal from jobs. Are you agree with this statement? Why or why not? Could you please share your experience? |
| 3 | Familial Factors | What are the family-related responsibilities which you think have adversely impacted your employment?<br>To what extent do you and your spouse/family members share domestic responsibilities? (For example, housework, shopping, cooking, washing, etc.)<br>How helpful do you think the family members would be if you were to run into difficulties at work? In terms of care for your matters and emotional support?<br>Do you think your job affected your family or marital life? How?<br>Have you experienced such feelings or incidents when you realize that your medical profession is affecting your children? Why and how?<br>Do you use any help or support of any kind in childcare arrangements? e.g., day-care centre, parents, tutor, maid, etc. Why?<br>Do you feel that your family responsibilities impact your job? Can you please share your experiences? |

hidden population i.e., non-working women doctors and gather unique insights by relying on referrals, social systems, and networks [39]. This sampling method facilitated the access to the women doctors through obtaining knowledge from individuals who may not be easily reachable through conventional sampling methods. In addition, personal contacts, and referrals were also effective to reach out non-working women doctors. These women doctors were contacted through various means such as phone, whatsapp and emails. Once contact was established, the researcher introduced themselves and the referrer, provided a brief overview of the research work, and sought permission to proceed with conducting an interview.

Before the interviews commenced, women doctors were assured that strict confidentiality would be maintained regarding their identity, and their data would be anonymized and solely used for research purposes. In addition, they were also informed that their interviews would be audio-recorded. However, it was emphasized that the audio file would be deleted once the investigation was concluded. Upon agreement, written consent from were acquired, with three women doctors granting oral consent. Subsequently, the interview time and location were determined based on the women doctors' ease and convenience.

The interview sessions commenced with greetings, and the researcher reiterated the purpose of the study. The probes were utilized with care to encourage in-depth conversation. The interviews were conducted in accordance with the women doctors' preferences, which included their residences, coffee shops, and public spaces. Typically, the conversations were held in both English and Urdu. All participating women doctors exhibited high proficiency in both English and Urdu languages, with Urdu being the national language spoken across Pakistan and English serving as the medium of instruction in medical degree programs. The data was then translated into English while retaining some quotes and terms in their original language. Naturally, each interview lasted between 50 and 90 minutes, and the data collection process was concluded in seven months approximately.

A total of 59 semi-structured in-person interviews were conducted across the Pakistan including Azad Jammu & Kashmir (AJK). Data collection and sampling was pursued until data saturation at which point where non new information emerged from the from the women doctors' narratives. However, two additional interviews were conducted as a precautionary measure to ensure that no new themes emerged from the data [27] which was not included in this study.

It is noteworthy that the four provinces of Pakistan exhibit distinct characteristics, encompassing variations in state procedures, cultural practices, and sociological attributes. Within these provinces, the perceptions of women's employment display diversity. Punjab, as the most populous province, is widely acknowledged for its hospitable nature and appreciation of music and arts. Furthermore, there is a growing acceptance and support for women's participation in the workforce. Sindh, home to the bustling city of Karachi, showcases a heterogeneous blend of urban and rural communities and demonstrates a relatively progressive mindset concerning women's employment. In Khyber Pakhtunkhwa (KPK), renowned for its awe-inspiring mountainous landscapes, traditional norms present challenges, although attitudes are gradually evolving. Baluchistan, the largest province in terms of land area, boasts expansive deserts, rugged mountains, and coastal splendor. The Baloch people, with a strong tribal identity, possess a rich cultural heritage. However, the perception of women's employment in Baluchistan tends to be more traditional and restrictive compared to other provinces. Notably, AJK situated in the majestic Himalayan and Karakoram Mountain ranges and recognized as a prominent tourist destination, exhibits a progressive mindset regarding women's employment.

Given the diverse characteristics of the provinces and the widespread occurrence of the research problem throughout the country, collecting data from the on a single city or province

**Table 2. Details of interview and locations.**

| Interview Type | Province | Cities | Number of Interviews Conducted | |
|---|---|---|---|---|
| | | | Never Practiced Medicine since graduation | Formerly employed |
| **59**- Semi structured Interviews | Sindh (25 Interviews) | Sukkur | 3 | 3 |
| | | Nawab-Shah | 2 | 3 |
| | | Larkana | 2 | 1 |
| | | Khairpur | 2 | 2 |
| | | Karachi | 3 | 4 |
| | Punjab (22 Interview) | Raheem-Yar-Khan | 1 | 1 |
| | | Bahawalpur | 1 | 4 |
| | | Multan | 2 | 5 |
| | | Lahore | 3 | 5 |
| | Khyber Pakhtunkhwa (2 Interviews) | Peshawar | 1 | 1 |
| | Baluchistan (2 Interviews) | Quetta | 1 | 1 |
| | Federal Capital (6 Interviews) | Islamabad | 3 | 3 |
| | AJK (2 Interviews) | Mirpur | 1 | 1 |
| | Total Interviews (59) | | 25 | 34 |

Source: Authors

could lead to inconsistent and heterogeneous experiences. Hence, it was essential to physically visit and document the extensive and varied experiences and perspectives of women doctors in their authentic environments, to thoroughly explore the research problem. Table 2 provides an overview of the number of interviews conducted in different cities across Pakistan.

Table 2 shows a total of 59 interviews were carried out throughout Pakistan. Out of these, 25 interviews took place in Sindh, 22 in Punjab, and 2 interviews each in AJK, Baluchistan, and Khyber Pakhtunkhwa (KPK). The table further reveals that among the 59 female doctors interviewed, 25 had never practiced medicine after completing their graduation, while the remaining 34 had previous work experience in hospitals.

## 2.5. Data analysis

The data was analysed through thematic analysis (TA) [40]. According to studies i.e., [25, 28], TA is compatible with the case study research design. The process of TA involves identification of the description content and then convert it to meaningful units through extraction. Thus, domains/themes of units (codes) were emerged from the women doctors' narratives of their experiences [27, 40]. Each interview, conducted in Urdu, was transcribed, and translated into English. In addition, field notes and the collection of documentary evidence also took place simultaneously. Prior to the first transcript being put into ATLAS.ti 22, all transcripts were examined twice. By allocating a single task to two researchers, the researchers attempted to avoid subjective biases to validate the findings. This procedure was performed independently for interviews and analyses. The analyses of two researchers were matched for internal validation (congruity). The remaining co-authors assessed the generated themes to ensure that they accurately reflected the content of the interviews until a consensus was established among all designated members of the research team.

## 2.6. Permissions and ethics approval

All women doctors signed a written consent form [41] except for three who exhibited reluctances in physically signing the written consent form prior to the interviews. Nevertheless,

they expressed their willingness to provide oral consent instead. The study was approved by the Ethical Review Board (ERB), Office of Research, Innovation, and Commercialization (ORIC), Sukkur IBA University, Pakistan.

### 2.7. Rigour

Standards for Reporting Qualitative Research (SRQR) were followed to ensure data analysis rigor [42]. Furthermore, research trustworthiness was examined [43]. Case study designs typically extensively use strategies for beefing up their level of rigor [44, 45]. Table 3 presents the specific criteria that were employed to ensure the trustworthiness of the present study.

## 3. Results and findings

### 3.1. Participants

A total of 69 women doctors were approached and screened for set-inclusive criteria. Out of 69 women doctors, three did not meet the inclusion criteria. Of 67 remaining eligible women doctors, two declined to participate in the study. Finally, 66 women doctors agreed to participate in the study. However, during the 59th interview, researchers reached the point of saturation. Table 4 shows the women doctors characteristics.

Table 4 shows among 59 women doctors, 34 was formerly employed in the hospitals whereas 25 had never practice medicine following graduation. Among 34 formerly employed women doctors, 23 were associated with public sectors; the other 11 were in the private sector. In addition, there was 17 single, 36 married women doctors whereas 6 were divorced.

## 4. Findings

The study revealed that most women doctors expressed a strong commitment to resuming their professional careers and making valuable contributions to the healthcare sector. However, they identified several factors that pose potential obstacles into employment. Thematic analysis of the interviews led to the emergence of three major categories: workplace challenges, socio-cultural obstructions, and family-related restrictions potentially hinder women's pursuit of medical practice in hospitals. These overarching themes were subsequently subdivided into more specific sub-themes are presented in Table 5 and the integrated framework is presented in Fig 1.

**Table 3. Research trustworthiness.**

| Criteria | Techniques Performed |
|---|---|
| Credibility | **Investigator triangulation**: Three researchers participated in the analysis of the interviews. The analyses were compared in subsequent team meetings to establish categories.<br><br>**Triangulation of data collection methods and sources**: Both semi-structured interviewers and documentary evidence were maintained. To further confirm the reliability of the results, interviews were also performed with two samples of the study's intended audience. |
| Transferability | **In-depth explanations of the processes used in the research**, including information on the characteristics of the researchers, participants, and contexts, the sampling techniques used, and the methods employed for data collection and analysis. |
| Dependability | **An external researcher audited the research guide**, concentrating on elements relating to the methodology and study design. In addition, the primary themes, coded verbatim quotes, and descriptions were also reviewed and confirmed. |
| Conformability | Investigator, participant, and data collection triangulation, the researcher's reflexivity was demonstrated by the execution of reflexive reports and the explanation of the study's rationale. |

**Table 4. Participating women doctors profile.**

|  | Participants Characteristics | Frequency |
|---|---|---|
|  | **Employment Status** |  |
| 1. | Women Doctors formally Employed | **34** |
|  | a. Public Hospitals | 23 |
|  | b. Private Hospitals | 11 |
| 2. | Women Doctors who never Joined hospitals | **25** |
| 3. | **Age Group** |  |
|  | 25–30 | 14 |
|  | 31–40 | 16 |
|  | 41–50 | 15 |
|  | >50 | 14 |
|  | **Marital Status** |  |
|  | Single | 17 |
|  | Married | 36 |
|  | Divorced | 06 |

## Theme 1: Workplace challenges

### 4.1. Ineffective recruitment process at government hospitals

The study found that every successful medical graduate with an MBBS degree and PMC service registration must undergo an additional examination to secure their jobs in government health institutions. In doing so, a state-owned agency named " Pakistan Federal Service Commission" (PFPSC) is responsible for fulfilling the hiring needs of different sectors, including health. Structurally, every province has its public service commission under the umbrella of the federal service commission.

**Table 5. Explored themes.**

| Explored Themes | Categories |
|---|---|
| 4.1. Ineffective Recruitment in Hospitals. | **Workplace Challenges** |
| 4.1.1 Seeking a Job in Private Hospitals. |  |
| 4.2. Improper Job Placement and Job Transfer Constraints. |  |
| 4.3 Insufficient Remuneration with Excessive Workload to Continue the Profession. |  |
| 4.4. Harassment in Hospitals |  |
| 4.5. Manifestations of Gender Roles Restricting Women Doctors to Practice. | **Socio-Cultural Obstructions** |
| 4.6 Hijab Restrictions. |  |
| 4.7. Marriage Incapacitating Women Doctors' Practice in Pakistan. |  |
| 4.7.1. Husband and In-Laws Restrictions. |  |
| 4.8. Familial Interference | **Familial Restrictions** |
| 4.9. Domestic Chores and Responsibilities. |  |
| 4.10 Childbearing and Caring. |  |
| 4.10.1. Day Care Centers- A Societal Dilemma in Pakistan. |  |
| 4.11. Family Roles that Make Women Exit from the Workplace. |  |
| 4.11.1. Patient Care Compromised |  |
| 4.11.2.Neglected Children |  |
| 4.11.3. Depressed Family |  |
| 4.11.4. Marital Relationship Endangered |  |
| 4.11.5.Women Doctors Physical and Mental Health Issues. |  |

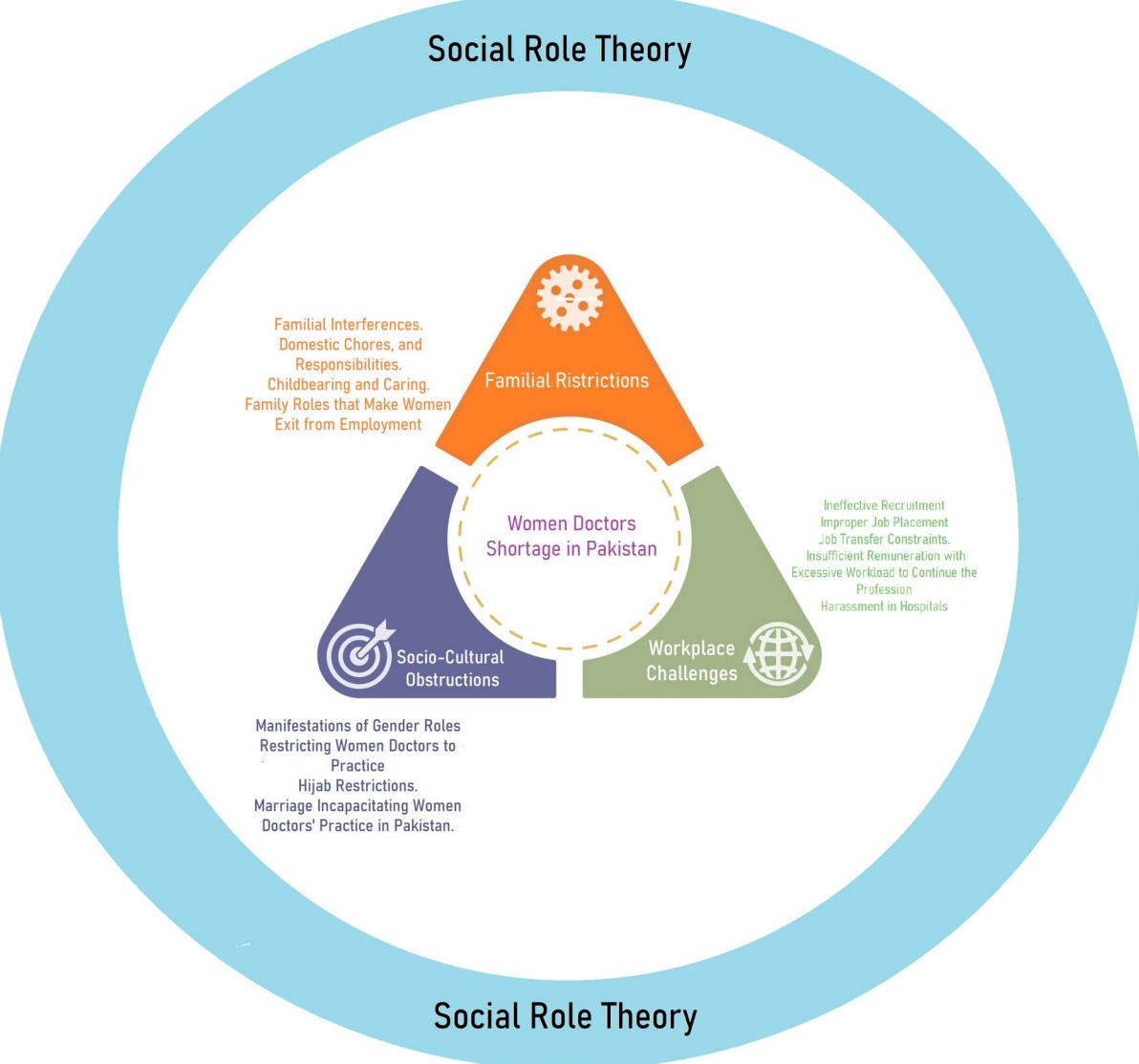

**Fig 1. Theoretical framework.**

The women doctors who were formerly applicants or associated with state-run hospitals were unhappy with the job recruitment process, especially with the PSC inductions system. The women doctors asserted that the appointment process is obsolete and full of weaknesses, preventing the entry of numerous medical graduates into public hospitals. In this context, the study explored two significant deficiencies obstructing many women doctors' entry to state-run hospitals. *1. No Hiring Despite Doctors' Shortage*: women doctors revealed that several medical officers (MO) positions are vacant in hospitals. However, the state authorities intentionally delayed or did not initiate the recruitment process despite various calls to secretariats and creating a massive shortage.

I would say a bunch of incompetent people with lot of mismanagement. They have no idea how few doctors deal with many patients daily with limited facilities. Our Medical

Superintendents (MS) of hospital repeatedly write to health secretariate for new placements, but no response since almost a year

(PQTA1, 31 to 40 years, married).

In the same context, another explored factor is *2. PSC Examination Comes Every 4th Year*: The present study found that medical doctors must wait at least 2–4 years for a job opening and then appear in the PSC induction examination:

There is a noteworthy and obvious flaw in the appointment process as many graduates wait for their commission examination before joining government hospitals. For this reason, most of the fresh graduates join private hospitals, which is an issue due to low salaries and excessive workload

(PLHE4, 31 to 40 years, married).

The study found that the hiring process is prolonged and takes approximately ten months to 1 year to complete.

I appeared in the exam in 2002 and got the offer at the end of December 2003. I did not join as I was getting married then, even though I worked in a private hospital for a year

(PISB3, 31 to 40 years, married).

In the meantime, many women doctors married and did not join the public hospital. Others might have left the country for good jobs and immediate responses.

**4.1.1. Private hospitals.**   The findings show that private hospitals are financially driven institutes owned by businesspeople and famous retired professors. In this way, they attract famous doctors of the field (preferably men) that bring more patients to their hospital, and women, on the other hand, suffer.

Private hospitals tend to offer a higher salary with countless perks to doctors with a good reputation in the area and a large social circle because these doctors eventually will bring business to the hospital. At the same time, women doctors have less societal exposure and do not bring business as much as men would bring due to a men-led society. Women are usually paid less

(PLAR1, 31 to 40 years, Single).

In the same connection, the research explored the private hospitals do not prefer married women, especially having children, for the job:

What is your plan for marriage? I think this question carries the weight of discrimination. Many of my fellows and I frequently encounter this question in job interviews, adversely affecting employment opportunities

(PLHE2, 41–50 years, married).

In addition, the women doctors discoursed that they believed that women with children consistently struggle with dual responsibilities, which may disrupt working. Likewise, married doctors and elderly children who wished to return to the hospitals also faced severe

discrimination. Unfortunately, the chances were far less for them as the government job has a specific age limit. Therefore, the only choice available is to join private hospitals:

> The hospital needs young and good-looking women for their patients and not aged like me. When women return to work, there is a bizarre mindset; they are not welcome. I feel I am already retired
>
> (PMUL1, 31 to 40 years, married).

In addition, considerable discrimination was claimed at a private hospital for hiring graduates from top medical colleges and universities:

> Hospitals like **** and *** only hire graduates from *** Universities. Among them, two private hospitals allow graduates from elite families.
>
> (PLAR1, 31 to 40 years, single).

## 4.2. Improper job placement and job transfer constraints

Job transfers were also among the main constraints that impeded many women doctors from working in hospitals. The study found that even after obtaining the job, the authorities issue posting order against the domicile and permanent residence certificate (PRC), which forces many women doctors to transfer their jobs according to their city of residence.

> Right after passing my commission examination, I was posted to Raheem Yar Khan City, which is almost 600km from my home (Lahore). My family and I were really disturbed and tried our best to get the proper placement. But they denied that the Lahore hospitals have no vacancies. My family did not allow me to live that far just for job and did not let me to join the hospital.

Also, the study found that few were utterly reluctant to work in developing areas such as towns because of poor security, no proper facilities for lodging, high chances of harassment, and strict tribal culture:

> I was posted in Umerkot, a small developing town in Sindh with more than 50 centigrade dry weather. There was no adequate guest house, hostel, or hotel. I was asked to live at a nurses' hostel, which I declined. I cannot imagine living one day over there. I never joined that place and applied for leave
>
> (PKHI1, 31 to 40 years, married).

While opting for the transfer, many women doctors termed the process as "Pathetic.". The process has various steps and hierarchies and requires the permission of the competent authority. Besides, the transfer process is reported as highly political, where nepotism and bribe culture are widespread:

> You cannot enter the Health Secretariat Building except you offer some money to a peon or gatekeeper. Now you can imagine how difficult it is
>
> (PKHP1, 41 to 50 years, married).

Women doctors expressed severe disappointment over the health department's role in dealing with this issue. The transfer process has been made complex and unnecessarily complicated, usually taking months to complete. Long waiting does not bring fruitful results, and many requests are mostly declined, or the provincial health secretariat keeps it pending for months for an unknown cause. Furthermore, the study found that not all the women doctors have enough finances to get their work done, nor do they have links with political and influential persons, and unfortunately, some women doctors had to sacrifice their jobs.

## 4.3. Insufficient rumneration with exessive workload to continue the profession

The women doctors show severe discontent with their salaries and other benefits in public and private hospitals. They stressed that there is no motivation or considerable gain to continue this profession. For that reason, most of them were influenced and convinced by family members to quit their jobs.

> I had to go through PSC exams twice, wait several grueling months in the secretariat offices, and almost beg for a job in my hometown, which I finally got, but that paid me absolutely nothing
>
> (PPESH2, 31 to 40 years Single).

The study found that women doctors (government employees) were paid Rs. 60,000 to Rs. 70,000 monthly salaries, which is minimal. Due to this issue, various protests/strikes, and complete lockdowns of OPDs occurred to get government agreement on the service structure. Similarly, the women doctors from private hospitals shows even more disappointment and anger over the remuneration paid. They revealed that private hospitals offer fixed and meager salaries even lower than public hospitals with no job security. The study found that every year a significant number of women doctors graduate from different medical colleges across the country, and the private hospitals take advantage of this surplus of graduates and compensate them on some terms which women doctors must agree to save their time:

> Private hospitals take advantage of fresh graduates and offer whatever they wish to pay, as they (private hospitals) know there are no jobs for fresh graduates in government hospitals. Pakistan produces more than 10,000 graduates annually, with a high proportion of women doctors. Do we have 10,000 jobs every year?
>
> (PISB3, 31 to 40 years, Married).

The women doctors expressed that the PMDC and health department also do not intervene in private hospitals' policies regarding their employees. Furthermore, few women doctors agreed that women lack negotiation skills and easily compromise on the terms and conditions.

## 4.4. Harassment in hospitals

The present study revealed that all women doctor experienced harassment in terms of verbal, psychological, and uncomfortable gestures, i.e., gaze, unwanted meetups, and unfavorable treatments at their respective hospitals. Severe concerns over the toxic culture of misogyny contaminating women's presence in the sacred profession were reported. The perpetrators

include men doctors' especially senior level in both government and private hospitals, patients' attendants, colleagues, and unknown people who used to stay on hospital premises, especially in government hospitals:

> Hundreds of doctors and women staff in medicine come across uncomfortable situations daily. We must be careful of every person, from the head doctor to the ward boy. I am not saying every person is a harasser. However, most of them are, and nobody takes notice as if it is a matter of honor
>
> (PLAR1, 31 to 40 years, single).

The study found "silence and ignorance" as a common way to deal with harassment. Women doctors briefed that many women doctors were scared that their complaint might create a "scene" and "scandal", which could damage their and family image, credibility, and reputation in the hospital and may force them to sacrifice their job. Alternatively, many women doctors see "transfers and resignations" as a curative measure. Surprisingly, the study found there is no official channel present to register women doctors who complain.

Resignation and transfers were the only safe options for the oppressed women doctors to deal with harassment issues because they feared their character would be assassinated and blamed for all the incidents. Besides, poor management essentially discourages women doctors from coming forward:

> You will be surprised to hear that government hospitals have no office to deal with this matter [harassment] because they were not interested in listening to these issues. Despite a significant number of women staffs, i.e., nurses, doctors, and administrative staff, little attention is paid because most women do not reveal their stories
>
> (PLHE2, 41–50 years, married).

Furthermore, the study found no written harassment policies, and women lack knowledge about their rights. The absence of countermeasures led to increased harassment incidents. Likewise, private hospitals have some departments to handle but have ineffective and unprofessional mechanisms to deal with this issue. In addition, the complaint is dismissed if the perpetrator relates to an influential person, political party, or charitable organization.

## Theme 2: Socio-cultural obstructions

### 4.5. Manifestations of gender roles restricting women doctors to practice

The study found that most women doctors were unhappy and pointed out patriarchal culture rooted in the country's societal system majorly hinders them from practicing medicine in hospitals. Specific to women's employment outside, the division of work is based on gender. In this context, most women doctors revealed that it is an inherent conviction in Pakistan that the men member of the family has the primary responsibility to earn money. On the other hand, women are not expected to work but fulfill domestic responsibilities such as looking after their husbands and children within the boundaries of the home. In this way, men have economic power, provide more control over women's lives, especially careers, and immediately place them in weaker positions. Non-conformity may lead to severe societal punishments where women are usually seen sacrificing their careers:

Women also have dreams, aspirations, and personal goals, and nobody is ready to realize and understand that. Society wants women to find and fulfill their dreams within the boundaries of home, which is unacceptable in any way

(PQTA2, 41 to 50 years, married).

Many women doctors wanted to work outside the home, but patriarchal norms suppressed their aspirations and choices. Most women doctors were battling to gain access to the freedom to live according to their wishes. In addition to contributing to income and fulfilling their personal goals, they were also ready to participate in household chores equally. The women doctors viewed the "sticky social norms" then led to the creation of different stereotypes again women". The study found medicine as a highly prestigious and appropriate profession for women in Pakistan, but the prevailing culture did not supersede the profession's nobility and restricted many women doctors. Society labeled their "earning" as morally and religiously objectionable to restrain women from working. In addition, the frequent mobility of women outside the home and contact with the opposite gender (strangers) in the hospital is considered a severe sin. Therefore, these behaviors are often perceived as "highly inappropriate" according to culture and religion:

Women who work outside the home are not nice ladies despite what they do [. . .]

(PSK2, 25 to 30 years, single).

The women doctors express that it is assumed that women having frequent contact with men, e.g., colleagues, staff, and patients or attendants, posit serious social, ethical, and religious issues on account of working women. However, the women doctors asserted that these stereotypes are irrational, illogical, self-created, and used to blame women and restrict their movements outside the home.

## 4.6. Hijab restrictions

Hijab/veil, or purdah as it is commonly called in Pakistan, is a system of gender-based ethics of interaction and a social institution that ensures modesty and propriety for Muslim women and men. From behavioral and interactional norms to sexuality, status, kinship, physical architecture of spaces and organization of work, hijab controls and encompasses almost every aspect of Muslim social life.

Under this umbrella, many women doctors discussed that they had faced many cultural and social difficulties in travelling to hospitals, which plays an essential role in their career outcomes. The findings revealed that the **frequent mobility of women** from their homes for work and interaction with men staff members is usually considered against the hijab. As a result, women face criticism and get labeled "not good women" by society:

In Pakistan, the hijab is used as a measurement tool for faith identification. For example, a woman having a burqa cover her head and face indicates she is high in faith and gets maximum respect. However, there is also a cultural and societal intervention in building that narrative

(PNS2, 31 to 40 years, single).

In addition, many women doctors experienced frustration and emotional tension using **public transport** while fulfilling the Hijab obligations:

After completing my duty, I had to use public transport such as a taxi or rickshaw, and bus, which is a horrible experience in Pakistan for any woman. Every time I observe the driver, conductor, and passengers staring at me or giving non-verbal gestures. It is a shame

(PISB3, 31 to 40 years, Married).

The women doctors reported it is tougher to come back at night from work without any men companion. It was very stressful for them because it violates the hijab and puts women at risk of harassment. The women doctors felt insecure and vulnerable because the societal structures in Pakistan put women at a disadvantage, more specifically, women from the middle class rather than women from more privileged backgrounds who travel on personal cars with a trustworthy man (husband, father, brother, or even a driver) or less privileged backgrounds who come back by walk from neighbouring farms in groups.

In addition to mobility, women doctors revealed that their mobility is also under **family surveillance**. The study found that extensive social networks and cell phones were used to extend the parental mobility of their daughters. Many women doctors declared that they were expected to seek their families' permission before leaving their homes or residence and inform them about their safe arrival at their destination. Married women doctors must take permission from their husbands (and parents-in-law if they live with them), even when visiting their parents' families.

## 4.7. Marriage incapacitating women doctors practice in Pakistan

Marriage was explored as one of the major causes and severe obstruction in women doctors' careers and employment decisions. In Pakistan, women need men and social support in every sphere of life. Marriage is considered a socio-cultural and religious obligation necessary to live a respectable life in the collectivist society of Pakistan. Marriage provides women with physical, emotional, and social protection to live their lives in a reputable way. Women doctors' responses indicated that parents and society become highly sensitive regarding women's marriage. When women doctors were asked more about this phenomenon, the majority explained that marriage is considered a mandatory part of life in Pakistan to have a respectable life. It is a "collective decision" or called "family decision" where everyone is usually involved in it, including the elders of the immediate family and close relatives. Marital decisions largely depend on familial norms, social class, case system, and geographical location. Another critical factor in everyone's mind is the "age factor." The women doctors explained that it has become complicated for a woman to get married if she has reached the age of 30 and above as she will not find a suitable match and will be considered over-age. Thus, it would be challenging for them to adjust to a new family.

In Pakistan, the preferred age for females to get married is between 20 and 26, especially among elite or educated middle-class families. Moreover, a solid societal narrative portrays this age range as the prime time for a girl to receive good marriage proposals. It is widely believed that if a girl exceeds this imaginary age limit set by society, she may struggle to find a suitable match. (PKHI2, aged 51 to 60, married)

Therefore, parents rush to get their daughters married rather than allowing them to pursue their careers. In addition to age, the research explored societal influences and women doctors' intention to settle down and have a good life partner at a suitable age. The women doctors view themselves as "victim" as they strongly assume that their professional lives were largely compromised after marriage.

I am not against the marriage. Not at all. It is our religious and cultural norms. However, I am afraid of the after-effects of marriage, for example, associated societal norms and family conditions which are man-made. These things are disastrous for any female working on the planet.

(PISB2, 31 to 40 years, Single)

Other women doctor narrated:

There are many unmarried women doctors working here either because they are over-age or divorced. When a woman prioritizes a job that is not her obligation, she loses her chance of getting married

(PLHE3, 31 to 40 years, Single).

In addition, the study found an astonishing notion that **"Doctor Brides"** are always in high demand in Pakistan, where mothers are trying to bring a doctor bride to their homes. Therefore, the medical degree is considered a "hot ticket" for women and parents to get a better marriage proposal; they become "trophy wives."

I believe it is purely a social status kind of thing, where people proudly introduce their daughter-in-law as a doctor. But in the end, they need the same doctor to make food for them and take care of the home's chores (PLHE4, 31 to 40 years, married)

It is clear from the findings that it is much easier for women to get married once they are doctors. However, it is not clear yet that either graduated women doctors have no genuine desire to serve as doctors or they use a medical degree to get a good marriage proposal, or the marital life really obstructs their career path and holds them back from practice in hospitals. Thus, this makes social discourse complex and challenging to understand.

**4.7.1. Husband and in-laws restrictions.**   The study found that the decision to allow women doctors to practice medicine solely relied on the goodwill and support of husbands and in-laws. Many women doctors had to quit their jobs because their husbands and in-laws did not allow them to work. Similarly, disobeying husbands or in-laws creates a highly unfavorable situation for married women in Pakistan that they do not want. One women doctor revealed her struggle to continue the work and said:

I resisted a lot initially, but my opposition made things worse, and finally, I was in a situation where I had to choose either marriage or a job. I sacrificed my profession for marriage. Anyone would do the same. Even my family and relatives admired my decision and acknowledged me as a sensible girl (PLHE4, 31 to 40 years, married).

Women doctors who convinced their husbands and joined the hospitals complained about their in-laws for deliberately exacerbating household responsibilities that made them eventually unable to practice medicine.

## Theme 3: Family restrictions

### 4.8. Familial interferences

The present study found that the familial structure grounds the unequal division of work within the family and consequently becomes a source of severe "deadlock" where many women doctors have observed sacrificing their professional careers while battling domestic and caring responsibilities. The study found that most family decisions are controlled and regulated by the father-in-law after marriage. In his absence, his mothers-in-law often lead the family. In this case, domestic chores such as cooking, cleaning, and washing were assumed as the primary responsibility of the daughter-in-law:

[. . .] In our family, women are not allowed to work; my role as a girl who looks after the family [. . .].

(PQTA2, 41–50 years, married).

Similarly, a few women doctors revealed that in-laws forced them to resign from their job. Therefore, most women doctors regarded the nuclear family system as favourable:

I have experience with the joint-family system; when you go to work, they have so many issues, like she is not giving much time at home, has a very irregular routine, even she does not cook, and does not look after her husband. If you live separately, you do not listen to all this

(PSK1, 31 to 40 years, married).

The women doctors revealed that the mother-in-law used to be dominant in the home. Typically, they expect their daughters-in-law to do domestic work, want to penalize them, and sometimes unnecessarily complain to their sons. Based on many women doctors' expressions, their mothers-in-law behaved as a "villainous character." The married women doctors had a "terrible" experience in a joint-family system in their personal and professional lives. The dictatorship of in-laws for gender ideology and their unnecessary interference often became a reason for married couples to settle in separate homes. However, the decision to live alone is not simple and easy. The women doctors disclosed that the couples must go through a series of battles to win:

I agree that living in a nuclear family brings peace; you are out of that mess. Nevertheless, do you think moving from a joint family is that much easy? If yes, then I bet every second girl would be living separately

(PQTA1, 31 to 40 years, married).

### 4.9. Domestic chores and responsibilities

The present study found that most women doctor were solely responsible for fulfilling household chores that make it difficult rather terrible for them to leave home for work.

It is a big clash that never ends in women's lives. On one side, you have your husband, home, and children; on the other, you have hospitals, patients, personal goals, and earnings. You must be 100 percent, as there is no margin of error on both side. Unfortunately, many struggling women have lost their battles and given up on their families

(PSK3, 31 to 40 years, married).

The women doctors indicated various household responsibilities that do restrict them from practicing medicine. In addition, nearly all women doctors revealed that men never share any domestic responsibilities, as Pakistani men are not expected to work in the kitchen or home whether they are husbands, brothers or sons, and therefore women have to perform all household chores:

My husband is also a doctor, and I used to fight with him because when we both came from the hospital, he lay down and watched TV, stretching out, and I had to go directly to the

kitchen to make food. Also, if there is any issue related to kids, then I must withdraw from work. Why do always we (Women) sacrifice?

(PLHE2, 41–50 years, married).

The research found that husbands involved in domestic work or helping their wives with different chores were labeled as "weak" and "slaves of a wife," and their role would be unacceptable for society.

## 4.10. Childbearing and caring

The current study found that the responsibility of childcare with daily domestic chores occupied many women doctor round the clock. Giving full responsibility to the mother, society neglects important and significant roles of the father in child nourishment. When inquired about the contribution of in-laws and other family members in their professional careers, majority of women doctors were uncertain in their responses and reported that just like domestic chores and husbands never took any part in childcare at home:

I have no time for myself, and no proper sleep. When my kids wake up and cry at night, I must take them outside the room so my husband can sleep well as he must go to work. I do not have time for my breakfast in the morning, as I am busy making for others. It is tough. I am exhausted with this routine of doing all alone everything

(PKHI1, 31 to 40 years, married).

The women doctors asserted that they must devote their time and energy to their children. However, they need the necessary support and equal participation from their husbands in domestic chores and childcare responsibility so that they can fulfil their career goals.

**4.10.1. Day care centers- a societal dilemma in Pakistan.** The women doctors discussed that the decision to hand over children to daycare centers is under the strong influenced of socio-cultural, familial, religious, and personal reasons. Above all, the women doctor were concerned about their child safety at daycare centers.

I am always afraid for my daughter's safety. You see, so many cases of child abuse are reported in Pakistan, and it is increasing daily. Also, what if they did not teach and entertain my daughter well or if she did not respect me as her mother maybe she will not behave well and become aggressive. These questions always come to my mind when I think of daycare centers

(PLHE4, 31 to 40 years, married).

The study found few women doctors who spoke in favor of daycare centers but did not find an appropriate and affordable place for their children:

Good daycare centers are scarce. It can only be found in metropolitan cities in selected areas. They are charging PKR 15,000 to PKR 20,000 for 8 hours and PKR 8,000 to PKR 10,000 for 5 hours roughly. I simply cannot afford this amount. Our total income is around 60,000, and how come we spend this vast amount?

(PSK3, 31 to 40 years, married).

### 4.11. Family roles that make women exit from the workplace

Various consequences experienced by women doctors that drove them to stay out of work are discussed as follows:

**4.11.1. Patient care compromised.**   All women doctors acknowledged that medicine is a stressful and sensitive profession with long working hours, dealing with emergencies, needing total mental concentration for proper diagnosis, and emotional stability to deal with complex patients. Due to tiring domestic responsibilities, it becomes difficult to concentrate on work that adversely affects patients' healthcare delivery:

> I still remember it was the darkest day of my life. I had an issue with my in-laws and my husband. Later, when I came on duty, I was stressed out. I had to work in OT that day, and what happened. . .. I gave the wrong injection to a patient. Luckily it was just a painkiller. I was given off for two weeks
>
> (PLHE2, 31 to 40 years, married).

Besides, the sickness of any family member, mainly their children, was additional pressure for the women doctors, which again had negative aftermaths on work.

**4.11.2. Neglected children.**   The women doctors were concerned that their children were neglected because of their hospital work. In addition, they expressed that they usually suffer emotional and psychological conflict. It was embarrassing when their children complained about their absence:

> I was speechless when my five-year-old daughter repeatedly asked me to leave the job and stay home. She usually texts me that she missed me even when I was doing house-job
>
> (PLHE2, 31 to 40 years, married).

**4.11.3. Depressed family.**   The majority of women doctors reported that they were depressed because they were feeling detachment from the family as they could not manage enough family time, which guilt them inside:

> When I got home, most of the time, I was exhausted. I know my family does not care about my job, so when I discussed my job-related issues with them, they told me, "*Why don't you leave*." That is why I started to keep my problems to myself and gradually felt isolated and disconnected
>
> (PKHI3, 31–40 years, Single).

Due to long working hours and emergency duties, women doctors could not actively participate in their families' meetings, social gatherings, and taking care of the family members, which eventually made them feel depressed most of the time. Similarly, the collectivist society of Pakistan and the relevant teachings of religion, i.e., Islam, direct them to look after their parents when they are ill. women doctors considered themselves as not "good daughters, "good sisters," and "good daughters-in-law" and received many societal reactions that influenced adversely on their lives.

**4.11.4. Marital relationship endangered.**   Many women doctors expressed that due to their work engagements at hospitals, they had less time for their homes and family, leading to marital conflicts. The findings show that the primary cause of the marital conflict was women

doctors' absence from home. As what husbands need when they are back home, their wives should be there for them and serve food to them.

> My husband wants me at home all the time. When I joined the hospital nearby, he was not happy. He was irritated all the time. He sometimes teased me that I do not focus on home and children
>
> (PMUL1, 31 to 40 years, married).

The women doctorrevealed that woman who work outside the home do not reflect the historical image of Pakistani wives. Historical images define a good wife as should "stay-at-home" who is sole purpose is to serve the husband. Also, she must wear jewellery and dress up nicely to please her husband. However, the women who work outside may not have time for dressing up or personal care, which disappoints their husbands' expectations. Most p women doctors believed that their husbands wanted to see them waiting at home; otherwise expected their life to be turbulent.

**4.11.5. Women doctors physical and mental health issues.** The women doctors were battling with long working hours, excessive workloads, and unsatisfactory personal life with no support, which put them in severe depression resulting in compromising health. They were upset as they did not have time for relaxation; dual responsibilities had shrunk their personal lives, and they had little time for themselves.

> Every day I thought of quitting. I did not have the stamina to get off the bed in the morning to start my routine, do all the chores, get ready, arrive at the hospital, and then go back home and work there. I do not remember how many painkillers I took daily. My health went down daily, but who cares
>
> (PQTA2, 41–50 years, married).

The women doctors discussed that they could not withstand performing dual careers and started developing severe health-related issues such as depression, anxiety, emotional instability, and continuous body ache.

These findings are visually presented in form of theoretical framework and depicted in Fig 1.

# 5. Discussion of the findings

## 5.1. Workplace constraints

The findings show that most women doctor were disappointed about their workplace experiences with human resource functions. Kebene [46] defines human resources in healthcare as "*different kinds of clinical and non-clinical staff responsible for public and individual health intervention*" that indicate and make healthcare workers responsible for overall health service delivery in the country. It is also to understand that human resource management (HRM) in health is more complex than regular HRM due to some unique characteristics. For example, the workforce is large, diverse, highly professional, and skilled occupations and powerful professional trade unions. Also, field-specific skills required in health sectors, fixed entry requirements, and standards usually controlled by the authorities (e.g., State, health department, PMC) make it more complicated to deal with them.

The findings revealed that poor governmental recruitment and appointment processes, transfer constraints, meager salaries with unjustified workload, and poor hospital management

system that includes harassment are the causes of women doctors' leaving or not joining the profession. These findings align with previous studies, e.g., [3, 21, 47]. Specifically, to doctors' shortage in hospitals. The women doctors indicated that the government's slow process and corruption for not appointing many doctors in hospitals despite the shortage, due to which patients face many chronic problems in seeking medical care in hospitals (especially government) throughout the country. Such reasons push patients into private hospitals, costing them hefty sums for their health care. The complaints of women doctors are not much different from those addressed by Habib et al. [48], who indicate that Government hospitals are overcrowded; thus, it has been challenging for doctors to provide patients with adequate care.

Considering Pakistan's health care, Hossain et al. [49] found that the government strictly centralizes the structure of human resource decisions, and there is no or minimal role of public health care institutes in the primary human resource function for staff. Moreover, medical doctors lack basic knowledge of their employment policies and procedures. In addition, gender notions are largely neglected in the process [49]. On the contrary, working in private hospitals raised different aspects of issues. Doctors are either hired through a referral system or from top medical universities. Besides, the selection is made over the preference of the marital status of women. For example, unmarried women and one who do not have plans to get married shortly have higher chances of getting a job than those who are already married and have children due to work-life imbalance. Nepotism in recruitment and selection remains a persistent challenge in societies. Owing to the accelerated professionalization of hospitals, it is now widely assumed that the management will base recruiting decisions exclusively on the skills and merits of applicants. Nepotism is widely viewed as immoral due to its correlation with some adverse side effects, including bias, corruption, lack of productivity, reduced work satisfaction, increased fatigue, and stress [50]. Ultimately, this creates a strong attitude that appreciation comes only from being a favorite of the supervisors. Sarwar and Imran [51] and Nadeem et al. [52] are also of the view that recruiting candidates based on favoritism "Sifarish" has become a common trend in Pakistan's workforce [51], especially in private organizations [52].

In addition, salary dissatisfaction was an essential factor raised in the present study. The women doctor were unhappy with their remuneration packages and found their salaries much less compared to the services they provide and the needs of their families. Low pay and insufficient benefits contribute to distress, turnover, absentmindedness, and poor performance among healthcare workers [53, 54]. Earlier studies in Karachi and Faisalabad have also observed concerns about low salaries among doctors [55, 56] and consider it one of the main reasons for dissatisfaction among them. Also, Yaseen [57] reported compensation as a significant factor in women doctor withdrawal from the hospital because there is not much motivation to continue the profession.

The findings show that the women doctors raised severe reservations about the integrity of the health department, particularly in dealing job transfer requests. They mentioned the irresponsive attitude of the officials at the respective health secretariat to resolving the real issues. It was observed that women doctors preferred to be near their residences due to family life, and Section 10 of the service rule also states clearly that civil servants may serve in their provinces based on permanent residence certificates and domicile. On the contrary, the rule has not been exercised moderately as Shah et al. [58] report that the doctors believed such rules are often bypassed and ignored, which heavily affects women doctors' employment decisions.

Finally, women doctors expressed the most profound concern for insufficient security provided at hospitals, which made them feel vulnerable and insecure. This violent behavior of patients is mainly caused by long waiting times and unsatisfied treatment. For instance, in some cases, patients and attendants allegedly accuse doctors of not treating them well and threaten to harm doctors [59]. Such incidents are often seen in emergency departments, and

earlier researchers [60, 61] have proposed the same findings. Not so long ago, sexual harassment in Pakistan was not considered seriously until the Protection Against Harassment of Women at Workplace Bill was passed in 2010, whose purpose was to provide safety to women at work. Unfortunately, despite this law, women still face sexual abuse throughout the country [61]. Patients and their families are the most cited sources of insulting or violent conduct.

Nevertheless, women doctors have documented cases of victimization by other doctors and colleagues. The findings of Miedema et al [62]; Nagata- Kobayashi et al., [63] are also in line with the present study. Another common theme noticed among women doctor was critics of personal matters (marriage, dressing, others) from male colleagues, and lack of security facilities was the most critical factor for harassment in hospitals.

Based on our findings, women doctors expressed the most profound concern for insufficient security provided at hospitals which made them feel vulnerable and insecure. This violent behavior of patients is mainly caused by long waiting times and unsatisfied treatment. For instance, in some cases, patients and attendants allegedly accuse doctors of not treating them well and threaten to harm doctors [59]. Such incidents are often seen in emergency departments, and earlier researchers [60, 62] have proposed the same findings. Not so long ago, sexual harassment in Pakistan was not considered seriously until the Protection Against Harassment of Women at Workplace Bill was passed in 2010, whose purpose was to provide safety to women at work. Unfortunately, despite this law, women still face sexual abuse throughout the country [61]. Patients and their families are the most cited sources of insulting or violent conduct.

Nevertheless, women doctor have documented cases of victimization by other doctors and colleagues. The findings of Miedema et al., [62]; Nagata-Kobayashi et al., [63] are also in line with the present study. Another common theme noticed among women doctor was critics of personal matters (marriage, dressing, others) from male colleagues, and lack of security facilities was the most critical factor for harassment in hospitals. Gender stereotypes in the light of social role theory, gender inequality, and workplace segregation rooted in Pakistan's patriarchal cultures are also discussed [51]. Management problems have been at the frontline of the concerns of women doctor. The governing structure and administration are unsupportive and impassive if any complaint is made. Women doctors pointe to organizational policies on harassment in their narratives; however, the mechanism for applying these policies seems either poor or ineffective. They reported that the hospitals they served had no clear or formal protocol specified on harassment policies. The women doctors was familiar with harassment guidelines, but it was discovered when further questioned that there were no appropriate structured protocols for filing a report. Many women doctor indicated that their only option in the event of harassment is to inform their superiors. Considering all these situations, women doctors endure incredibly frustrating and embarrassing situations. Additionally, in most cases, all these delicate matters are not discussed with family and friends due to cultural norms, which lead women doctors to suffer in silence and ignorance, have themselves transferred, or resign from their work.

## 5.2. Sociocultural obstructions

The findings revealed that the socio-cultural system is based on strong patriarchal beliefs. The persistence of stereotypical solid gender roles and expectations was in everyday practice. The women doctors expressed their position in the family or society as the carriers of honor of their entire family, where they must act according to rules set by the socio-cultural system. These findings are in line with the study of Pleck [31], who indicated that social systems which is based on families and social beliefs set both sexes' roles, and non-compliance with societal

norms is considered unacceptable and punishable in restricting at-home, not allowing women to pursue education or employment [31, 64]. On the other hand, professional women are usually considered "Selfish" or "self-centered" and arrogant [64–66] and given less respect by society [64]. However, Reskin [67] and Pedavic [68] found that women are under pressure to get socially appropriate professional roles. The women doctors expressed that the prevailing socio-cultural system suppressed women at large. In this connection, Pakistan is one of the countries with a massive gender disparity [16]. In addition, the country is ranked 151 out of 153 in the Gender Gap Report, which shows that women suffer in almost all fields of life.

The present study also identified the "Societal-driven concept of the Hijab" that restricts women doctors' careers in various ways. Hijab is a partition or a barrier for women to keep them separated from strange men. The study found two schools of thought about hijab. First, wear a veil to cover the body. Second, physical barrier where women cannot interact with strange men. The women cited that their restricted movement outside the home due to societal expectations and safety considerations under the hijab significantly influenced their career decisions. The findings are consistent with the study of Fayyaz [69] and Qaisar [70], who stated that Pakistani society demands all respectable women to stay at home and only move outside when fully veiled and indistinct.

Furthermore, the women doctors reinforce that even though the hospital is deemed comfortable and appropriate, commuting to work may produce an exposure that contravenes social codes. For example, in the present research, women doctors were concerned about commuting to work by public transport, and it causes additional mental and emotional stress during night shifts. Another study conducted in Pakistan also mentioned commuting as one of the barriers to the academic careers of women [71]. Cultural constraints on night-time mobility are especially significant in Pakistan, particularly for women doctors, most of whom have night and evening shifts. Public places are viewed as dangerous during the night, so staying out at night or spending the night outside the home is considered disrespectful and leaves a negative impression on women and their families. Women doctors, therefore, prefer to choose specialties that may not have night shifts. Nevertheless, this does not seem very easy because the hospital administration cannot give all women doctors that favor.

The findings are consistent with studies conducted in Iran [72, 73], where women healthcare workers have mentioned negative socio-cultural and familial attitudes toward night shift working. A lack of transportation choices for women dramatically limits their lives and influences how they work and what sort of jobs they accept [74]. A survey of women traveling daily in Karachi reported that 85 percent of working women experienced harassment while traveling on public transport last year [75]. Women's involvement in the workforce is strongly related to car ownership in the household. This may indicate a relationship between income and the involvement of women in the labor force, and it may even impact mobility; women who own cars may drive more quickly to work without being abused or stigmatized [76]. Ahmad [59] also indicates a lack of affordable and accessible transport as one of the significant challenges to women's labor force participation.

Women doctors consider marriage an obstacle, not a transition in their lives. The study found that Pakistani society promotes arranged marriage philosophy where parents play a crucial role in deciding whom and when to marry. These findings confirm the exploration of Masood A. [77], who sees Pakistani marriages as a transaction to maximize the social capital of family and women doctors who become desired brides due to their cache of social, cultural, and economic capital. women doctors who did not attend the hospitals blamed their parents, who prioritized marriage over their jobs, while others listed their in-laws, especially husbands, who did not allow them to pursue employment. Moazam and Shekhani [8] have a similar finding: they stated that women are still marginalized in Pakistan and are trained to reflect family

honor. In this way, actions that contravene values and practices, such as refusing parents or husbands in some matters, are perceived as critical factors in honor crimes and other acts of violence [78]. Consequently, career routes for women are not as straightforward as those for men and are riddled with intermittent breaks and sudden desertions. Men are dealing with shifting family dynamics, whereas women often face the challenge of trying to choose between their jobs and their families, and the dual commitment of balancing family and work often leads them to quit their jobs [9, 79].

### 5.3. Familial interferences

The familial system of Pakistan is considered a product of the prevailing socio-cultural system in Pakistan. The findings show that living in a joint and extended family is a common culture in Pakistan. Rafique [80] and Iqbal and Bashir [66] have defined an extended or joint family as one in which all family members, i.e., parents, grandparents, siblings, their spouses, and their children, live together in the same house. Nevertheless, urbanization, socioeconomic tensions, and economic displacement gradually contribute to a move towards nuclear households, which has pros and cons [81] (Akhtar, 2019). After marriage for ages, it has been common practice in Pakistan for a girl to stay with her in-laws after marriage [77] (Masood, 2019). She puts her husband in a very high position in which she obeys him and is obligated to follow her in-laws. In the case of a ruling mother-in-law, she is supposed to do the household work and look after her children, irrespective of her working status. She is puzzled and unable to grasp who is the natural decision-making body is and who the household boss. Under such a situation, a woman finds it very difficult to balance work and family, which is when most women tend to settle on their occupations and employment. In a recent study in Karachi (Pakistan), Shaikh et al. [82] (2018) reported that more women than men were admitted to and graduated from medical colleges. However, most of them end up as housewives instead of practicing medicine, and marriage has been recognized as a primary variable in this burnout [83].

Moazam and Shekhani [8] agree that the failure to follow careers for many women seemed to be linked to husbands and in-laws. She explained that after marriage, women doctors are often refrained from working by their husbands or in-laws [82]. All these observations are the crux of the present study. Women in Pakistan are still deemed solely responsible for meeting children's and families demands. They continue to face societal pressures and are required to perform the conventional position of homemakers. Conflicts occur when the family, particularly the husband, is not prepared to compromise to ease pressure on women, which increases pressure among working women [9] and results in severe work-life conflicts.

In line with these explorations, Belliappa [84] discusses women's life when she needs their parents' support, and women try to add to their families' status and prestige through their success. Even though their career accomplishments are a source of honor and pride for the family, paradoxically, women's careers also put the family's prestige in danger in several ways. Firstly, families in the higher socioeconomic classes take pride in not using women's income for household expenditures. When women work, families make it a point to ensure that it is not perceived as the women contribute to the family income. Secondly, their interactions with male colleagues are also a source of slander for the family, which may cause some families to discourage women's careers. Thirdly, the possibility of neglect in domestic and maternal responsibilities is a significant consideration. While marriage and motherhood confer a sense of honor to their identities [85], married women are more likely to be criticized for neglecting family obligations [84]. In our case, it appears that women internalize social expectations through family expectations. When they find these expectations in contrast to their career aspirations, they are likely to put the family's expectations first to reciprocate the support they

receive from the family. The relational framework [86] explains this internalization of social protocol and its reflection through personal decision or choice. It explains the interplay between the overarching structural factors and individual agency.

The women doctors revealed several consequences due to the struggling lifestyle. These include compromising patient care in hospitals, neglecting children, depressing family life and marital relationships, and health problems. Several studies have studied possible causes of stress among doctors. Such results show a lack of control, difficulties managing personal and professional life, extensive administrative activities, and a high number of patients [87–89].

## 5.4. Theoretical contributions

The research has made significant theoretical contributions by employing Social Role Theory and Spill-Over Theory to examine the women doctor shortage phenomena in Pakistan. Through these theoretical lenses, findings revealed a complex interplay between societal expectations, gender roles, and familial dynamics that shape women's decisions and experiences in the medical profession. Specifically, from social role theory, the study has uncovered how traditional cultural values and entrenched gender roles in Pakistani society act as barriers, discouraging women from actively practicing medicine. This theoretical lens has provided valuable insights into the socio-cultural obstacles that hinder women's participation in the medical field, expanding our understanding of the complex dynamics within the healthcare sector. Additionally, the incorporation of spill-over theory extends the understandings on familial interferences on women doctors' professional engagement. By investigating the impact of factors such as hijab, marriages, domestic chores, the influence of in-laws, and patriarchal familial dynamics, the study has highlighted how these external pressures can significantly impact women doctors' decision-making and career trajectories.

Overall, these theoretical contributions have provided a critical lens through which to examine the multifaceted factors that discourage women from practicing medicine in Pakistani hospitals. By challenging existing societal norms, gender roles, and familial dynamics, the research has advanced the understanding of the barriers faced by women in the medical profession and opened avenues for future research and interventions aimed at promoting gender equality and empowering women within healthcare settings of Pakistan.

## 5.5. Practical contributions and recommendations

The ongoing debate in newspapers and social media often accuses women doctors of being responsible for the workforce shortage. Nevertheless, this study explored the causes that hinder women doctors from working in the healthcare system. Based on the findings, women medical students must realize that their contributions significantly mitigate the shortage issue and positively influence the overall healthcare system. In this way, medical schools in connection with PMC should keep track of graduated students and their parents, provide them with necessary employment information, and counsel them if they have any issues. In the hospital, strict policies against harassment and gender discrimination must be drawn, and the respective head of departments should be responsible for safeguarding decorum. Similar measures were taken at Harvard University and John Hopkins Departments of Medicine to improve the climate of their science, engineering, and medical departments for women [90]. Currently, there is no professional association of women doctors that addresses their issues, and there is no women representation in strategic and administrative policymaking. To counter this issue, a minimum number of women doctors should be present in designing new health-related policies and hiring and transfer committees. In addition, instead of using policies like central induction, the

need of underserved rural areas can be better served by providing monetary incentives for doctors willing to work in rural areas or making it a compulsory part of post-graduate training. These are the policies taken in the United States and Australia to counteract the shortage of doctors in underserved areas. Multidimensional intervention and cultural changes are needed to achieve gender equality at the root level, not just superficially. To this purpose, Pakistan's regulatory agencies (PMC) must regulate regulatory and legal activities and ensure transparency. Religious scholars must analyze political ideas and historical practices on women's labor to bring about positive change.

Many hospitals worldwide offer career and work flexibility for their workers with responsibilities (childcare). This policy is not available for Pakistani women doctors. Flexible policies in terms of training and working hours may be followed to promote women's employment rather than to give breaks during training. It is because women doctor often delay their careers or take breaks until their children start going to school. Giving relaxation in working hours will result in fewer breaks in their resume but will increase the number of practitioners in the health economy (for a detailed discussion of flexible work structure and its importance of persistence of women, see Stone [91]). Besides daycare centers, 24-hour care for children may be available for doctors who also have night shifts.

Moreover, issues pertaining to families do not support childcare. Therefore, childcare scholarships should sometimes be provided to offset the cost of at-home care. These facilities will help women and men, doctors, equally in balancing work-family responsibilities.

Another way to increase women doctors' participation is to rethink how medical labor should be organized. Some initiatives have already been taken to bring women doctors back into the workforce and provide health care in underserved areas using the internet and mobile technology. Sehat Kahani and DoctHERs in Pakistan create a virtual marketplace where women doctors can treat patients remotely [92]. Reviewing health care with fewer regulations and more personalized work structures could result in more affordable health care and an increased number of women doctors in Pakistan.

## 5.6. Study limitations

Interviews were conducted in metropolitan cities of Pakistan so there are chances that women doctors working in rural areas where gendered norms vary significantly may have different workplace experiences. Moreover, the socioeconomic status of women doctors may have influenced their responses. Almost all women doctors belong to the middle of the upper class. The author hypothesizes that women doctors from lower socioeconomic backgrounds would have different issues (perhaps because the medical profession is associated with significant status and economic mobility). The absence of lower socioeconomic class women from working indicates how vital socioeconomic class can be in determining access to educational facilities. This aspect is not covered in this research since it only focused on the women who managed to become doctors. The data were collected for seven months; it is impossible to ascertain whether the findings will stand true over a more extended period, say 10 or 20 years. This research has little consideration of the tribal and challenging areas regarding access to education, law and order situation, political situations, and strict tribal rules prevailed in these areas. In addition, there are some inescapable limitations associated with interviewing in qualitative research, such as taking what an interviewee says at face value, the risk of influencing, and being influenced by, women doctors, and focusing on thoughts and reconstructions of events at the expense of direct action [93–95]. However, the stated risks are always present, but considerable efforts were taken to minimize the effects on the findings.

### 5.7 Future avenues of research

The findings can be more refined if we widen the scope of data collection. For example, interviews with PMC officials, women doctors' parents, husbands, and other family members related to women doctors' employment and withdrawal decisions. Furthermore, the explored factors provide a basis for the quantitative study to testify it on a large population of non-working women doctors for generalized understanding. Future research can also examine the experiences of women still under-represented in other scientific fields, particularly engineering (which is considered "masculine work") in Pakistan. A similar study may be conducted to understand women's experiences in other educational institutions and organizations.

## Author Contributions

**Conceptualization:** Ali Raza, Junaimah Jauhar.

**Data curation:** Ali Raza, Ubedullah Memon, Sheema Matloob.

**Formal analysis:** Ali Raza, Noor Fareen Abdul Rahim, Ubedullah Memon.

**Funding acquisition:** Sheema Matloob.

**Investigation:** Ali Raza.

**Methodology:** Ali Raza, Junaimah Jauhar, Noor Fareen Abdul Rahim.

**Project administration:** Noor Fareen Abdul Rahim, Ubedullah Memon, Sheema Matloob.

**Resources:** Ali Raza, Noor Fareen Abdul Rahim, Ubedullah Memon.

**Software:** Ali Raza, Noor Fareen Abdul Rahim, Sheema Matloob.

**Supervision:** Junaimah Jauhar.

**Validation:** Ali Raza, Junaimah Jauhar.

**Visualization:** Ali Raza.

**Writing – original draft:** Ali Raza.

**Writing – review & editing:** Ali Raza, Junaimah Jauhar, Noor Fareen Abdul Rahim, Sheema Matloob.

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
