## [Decision Letter · Decision Letter 0]

12 Apr 2023

PONE-D-23-00010FACTORS LINKED WITH THE SHORTAGE OF WOMEN DOCTORS IN THE HEALTHCARE SYSTEM OF PAKISTANPLOS ONE

Dear Dr. Raza,

Thank you for submitting your manuscript to PLOS ONE. After careful consideration, we feel that it has merit but does not fully meet PLOS ONE’s publication criteria as it currently stands. Therefore, we invite you to submit a revised version of the manuscript that addresses the points raised during the review process.

Please also consider the following suggestions:1. The abstract states that 59 interviews where done, while 69 participants were mentioned in the method section. Rather state that 69 potential participants were recruited while 59 interviews were done utilized for this study. People that did not finally grant the interview for the study are not referred to as participants. 2. If in-depth interviews were really done, then interview guide (NOT PROTOCOL) should be the instrument used for data collection. The interview guide allows for free flow of information from the participant without following any protocol so that even relevant information mentioned by the interviewee, which the interviewer might not have anticipated before the interview can still be explored deeply. The interview guide will then be just a guide to enable in-depth interview. 3. In line 186, you referred to the instrument for data collection as "question guide." while it was referred to as 'interview protocol" in other places.  Consistency and following the suggestion in point number 2.4. .Consider removing table 2. It will be perfect to be in your PhD thesis. The mention of it in line 186 is sufficient for an article.5. Please first table was numbered "2.1" Please number the tables properly. Let it be tables 1, 2, 3 etc. 6. Table 2.2 referred to the instrument for data collection as "interview topics." Please did you do an semi-structured in-depth interview? Where are the probes that were used to illicit more information during the interview? Open ended leading questions with potential probes should be stated in the interview guide for semi-structured interviews.7. In line 198, the objective is not to find out why women does are out of practice, but rather why their is a shortage of women doctors...8. Line 2013, change assent to consent. Minors (those below the legal age t give consent) give assent9.Line 203 to 205, "Following their assent, participants were assured of their identity 204 and discussion confidentiality and informed that their conversation would be presented collectively, 205 i.e., a research report." You don't assure people of their identity. You assure them that you will maintain confidentiality, their data would be made anonymous and used only for research purposes. 10. Line 206 " Additionally, it was clear to them that the interview would be audio-recorded for later analysis." Rewrite this sentence to say that the participants were informed that the interview would be audio-taped/recorded.11. Did you obtain consent for the audio-recording from the participants? Was it a written consent? This is different from the consent for audio-recording.13.Line 107, What do you mean by "audio records would be erased as soon as the investigation ends?" When would you say the investigation has ended? What is the policy of the Human Research Committee of your institution concerning audio-recorded data? Please when submitting the revised version, attach the consent form used for the interview and the consent form used for audio-taping. Also provide a copy of the ethical approval certificate for this study.14. Line 219, how many participants were hesistent to have their interviews recorded? Did they consent to audio-recording before the interview began?15. Was the interview face to face? Who were present during the interview (e.g interviewer and interviewee only etc). As stated previously in the manuscript, did the interviewer travel to all the major cities in Pakistan to collect this data? Were all the participants fluent with Urdu and English?16. Now in line 221, you mentioned 59 participants again.17. Add citation for that definition of theoretical sampling. I think the definition given speaks more to "data saturation" method. Theoretical sampling involves collecting data and analysing simultaneously to determine what to collect next in order to develop a theory. Please clarify and state what you did.18. Line 229 -232, how did you decide on which cities to collect data since they all have different languages, culture, social life etc. Authors now mentioned that data was collected from ALL FOUR provinces  of Pakistan, whereas they previously stated that data was collected in MOST of the provinces in Pakistan. I would suggest you remove remove some words that might not be necessary (see line 210) and add some information about the cultural background of where the data was collected. This would help contextualize and enable the reader to have a better understanding of the study.19. Table 3.3.Some columns have 3 cities/provinces with 2 interviews (both categories made up the 2) done in them? Also, seperate the provinces from the cities to give a clear picture.20. The number of interviews reported in table 3.3 do not add up at all.21. Line 229, the quote do not match the description given above it. Please look out for this mistake all through the result section and correct it.22. Lines 479 to 491 seems to speak more to socio-economic and strict parental control as barriers, instead of the hijab restriction which was the theme for that section.23. Add quotes to support the information in theme 4.2.3. Look out for this all through the result section and fix it. The quotes are the data. 24. You can present a table to show the topics presented in the results and discussion section as themes and their subthemes. Better organization of the data will make it easier for the readers to follow through.25. Consider removing any information that does not speak to the topics discussed in the result and discussion. Also look through the paper and remove any unnecessary repetition of information.26. I do not think the two figures presented are relevant for publication in this paper. Also, did you create the figures ( as they speak about the government but have no reference beneath them)?

We look forward to receiving your revised manuscript.

Kind regards,

Oluchukwu Loveth Obiora, PhD

Academic Editor

PLOS ONE

Journal Requirements:

3. Please ensure that you refer to Figure 4.2 in your text as, if accepted, production will need this reference to link the reader to the figure.

4. Please upload a copy of Figure 4.3, to which you refer in your text on page 16. If the figure is no longer to be included as part of the submission please remove all reference to it within the text.

5. We note you have included a table to which you do not refer in the text of your manuscript. Please ensure that you refer to Table 3.7 in your text; if accepted, production will need this reference to link the reader to the Table.

Reviewers' comments:

Reviewer's Responses to Questions

**Comments to the Author**

1. Is the manuscript technically sound, and do the data support the conclusions?

Reviewer #1: Yes

Reviewer #2: Yes

2. Has the statistical analysis been performed appropriately and rigorously? 

Reviewer #1: Yes

Reviewer #2: N/A

3. Have the authors made all data underlying the findings in their manuscript fully available?

Reviewer #1: Yes

Reviewer #2: No

4. Is the manuscript presented in an intelligible fashion and written in standard English?

Reviewer #1: Yes

Reviewer #2: Yes

5. Review Comments to the Author

Reviewer #1: As a reviewer I am impressed with the extensiveness of the research and congratulate the researchers to come up with an impressive manuscript. However, I have few suggestions to consider.

1. As it is a qualitative study, manuscript title looks like a quantitative one. It should be more sympathetic to female doctors. such as constraint and inhibitory factors or like that.

2. Similarly, introduction section should be more polite towards female doctors and it should not make them culprits for shortage of doctors in the country.

For example "According to PMC (formerly PMDC), the foremost cause of the medical doctor shortage is non-practicing women doctors in Pakistan." I am sure it not the only cause of shortage of doctors in Pakistan. However, whole of the introduction section gives such impression.

Further, I believe researchers should not come up with the exact number (178,440) on shortage of doctors. As, countries with best healthcare systems are suffering with shortage of medical staff and achieving set standards on number of doctors is almost impossible. Further, we as researchers do not have authority to come-up with the number on shortage of staff.

3. Manuscript looks a bit lengthy. Introduction section can be reduced. In my view there is no need to provide BoDs in introduction as this not the area of interest in this research. Introduction section should be reduced to motivation of the study, goal, objectives and contribution.

Good luck.

Reviewer #2: Enjoyed reading the paper after a long time as it’s very well written, well supported with evidence and in the form of a story making the paper a very interesting piece for work for readers. This topic really needs to be researched and I am glad to see some work in this area as this area is being understudied. Minor corrections mentioned below:

Contribution needs to be clearly identified in the introduction and then in the findings. Currently, it looks like more descriptive research whereas there is a great potential in the paper to be repositioned as a theory building paper. Some hints given at the end.

Theoretical lens can be expanded. Although it’s a qualitative paper based on the data, but little bit addition can add value, but I can see a very long discussion at the end which can be summarized in a form of a table and supporting studies maybe added in the column next to the findings of the study and in this way. Because adding the theoretical part may lengthen the paper in case discussion is not trimmed or summarized.

Table 3.3 review number please, there seems discrepancy. Similarly, check total number across the paper and tally.

“I appeared in the exam in 2002 and got the offer at the end of December 2003. I did not join.

313 as I was getting married then, even though I worked in a private hospital for a year (PISB3,

314 31 to 40 years, single).” The participant is single or married above.

Findings 4.1 and 4.1.1.1 needs to be converted into themes. Resulting themes. Like 4.1.2. For example, in effective recruitment processes at the government level etc.

Explain Hijab as readers may not understand the concept and underpinning philosophy if not from the same socio-cultural and religious background.

The data is very rich, an integrated framework at the end of the findings will summarize the paper findings. Why factor needs to be clearly mentioned as its why that will be the theory development. Now what and how are there. Concepts and their relationships but why that exist will contribute the theory. At the moment the paper is descriptive with rich data. This comment goes back to the first comment that contribution of the paper needs to be highlighted and this is possible only in case of the development of the framework and theoretical prepositions derived from the framework.

Recommendations to improve the situation needs to be added as the paper can have practical solutions for the situation.

6. PLOS authors have the option to publish the peer review history of their article (what does this mean?). If published, this will include your full peer review and any attached files.

Reviewer #1: **Yes: **Mohammad Jamal Khan

Reviewer #2: **Yes: **Amira Khattak

---

## [Author Response · Author response to Decision Letter 0]

14 Jun 2023

Thank you for the valuable suggestions. All the comments regarding requested revisions are mentioned in separate file named Reviewers Response. However, the some minor requested edits were made in manuscript file. Thank you.

---

## [Editor Report · Decision Letter 1]

28 Jun 2023

Unveiling the Obstacles Encountered by Women Doctors in the Pakistani Healthcare System: A Qualitative Investigation

PONE-D-23-00010R1

Dear Dr. Raza,

We’re pleased to inform you that your manuscript has been judged scientifically suitable for publication and will be formally accepted for publication once it meets all outstanding technical requirements.

Kind regards,

Oluchukwu Loveth Obiora, PhD

Academic Editor

PLOS ONE
---

## [Editor Report · Acceptance letter]

5 Jul 2023

PONE-D-23-00010R1 

Unveiling the Obstacles Encountered by Women Doctors in the Pakistani Healthcare System: A Qualitative Investigation. 

Dear Dr. Raza:

I'm pleased to inform you that your manuscript has been deemed suitable for publication in PLOS ONE. Congratulations! Your manuscript is now with our production department. 

Kind regards, 

on behalf of

Dr. Oluchukwu Loveth Obiora 

Academic Editor

PLOS ONE